# Inhibition of Phytopathogenic and Beneficial Fungi Applying Silver Nanoparticles In Vitro

**DOI:** 10.3390/molecules27238147

**Published:** 2022-11-23

**Authors:** Ileana Vera-Reyes, Josué Altamirano-Hernández, Homero Reyes-de la Cruz, Carlos A. Granados-Echegoyen, Gerardo Loera-Alvarado, Abimael López-López, Luis A. Garcia-Cerda, Esperanza Loera-Alvarado

**Affiliations:** 1CONACYT-Centro de Investigación en Química Aplicada, Depto. de Biociencias y Agrotecnología. Blvd, Enrique Reyna H. 140, Saltillo C.P. 25294, Coahuila, Mexico; 2Instituto de Investigaciones Químico Biológicas, Universidad Michoacana de San Nicolás de Hidalgo, Avenida Francisco J. Múgica S/N Ciudad Universitaria, Morelia C.P. 58030, Michoacán, Mexico; 3CONACYT-Universidad Autónoma de Campeche, Centro de Estudios en Desarrollo Sustentable y Aprovechamiento de la Vida Silvestre (CEDESU), Av. Agustín Melgar, Colonia Buenavista, San Francisco de Campeche C.P. 24039, Campeche, Mexico; 4Colegio de Postgraduados, Campus San Luis Potosí, Innovación en Manejo de Recursos Naturales, Iturbide 73, Salinas de Hidalgo C.P. 78600, San Luis Potosí, Mexico; 5Tecnológico Nacional de México, Campus Instituto Tecnológico de la Zona Maya, Carretera Chetumal-Escárcena, Km. 21.5, Ejido Juan Sarabia C.P. 77965, Quintana Roo, Mexico; 6Centro de Investigación en Química Aplicada, Depto. Materiales Avanzados. Blvd, Enrique Reyna H. 140, San José de los Cerritos, Saltillo C.P. 25294, Coahuila, Mexico; 7CONACYT-Universidad Michoacana de San Nicolás de Hidalgo, Avenida Francisco J. Múgica S/N Ciudad Universitaria, Morelia C.P. 58030, Michoacán, Mexico

**Keywords:** nanotechnology, sustainable agriculture, phytopathogens

## Abstract

In the current research, our work measured the effect of silver nanoparticles (AgNP) synthesized from *Larrea tridentata* (Sessé and Moc. ex DC.) on the mycelial growth and morphological changes in mycelia from different phytopathogenic and beneficial fungi. The assessment was conducted in Petri dishes, with Potato-Dextrose-Agar (PDA) as the culture medium; the AgNP concentrations used were 0, 60, 90, and 120 ppm. *Alternaria solani* and *Botrytis cinerea* showed the maximum growth inhibition at 60 ppm (70.76% and 51.75%). Likewise, *Macrophomina* spp. required 120 ppm of AgNP to achieve 65.43%, while *Fusarium oxisporum* was less susceptible, reaching an inhibition of 39.04% at the same concentration. The effect of silver nanoparticles was inconspicuous in *Pestalotia* spp., *Colletotrichum gloesporoides*, *Phytophthora cinnamomi*, *Beauveria bassiana*, *Metarhizium anisopliae*, and *Trichoderma viridae* fungi. The changes observed in the morphology of the fungi treated with nanoparticles were loss of definition, turgidity, and constriction sites that cause aggregations of mycelium, dispersion of spores, and reduced mycelium growth. AgNP could be a sustainable alternative to managing diseases caused by *Alternaria solani* and *Macrophomina* spp.

## 1. Introduction

Nanotechnology involves manipulating materials at length scales in the range of 1 to 100 nanometers, which, according to the National Office of Nanotechnology Coordination, is where the physical, chemical, and biological properties of nanoparticles allow novel applications and functions [1]. The development of nanotechnology and biotechnology (bionanotechnology) has become important for sustainable agriculture, which is considered one of the fields in which nanoparticles have greater application and growth potential [2]. This is because they help to develop environmentally friendly biosynthetic processes through the synthesis of nanomaterials [3]. Bionanotechnology can use biological sources to synthesize a large number of nanoparticles of different shapes and sizes [4] for specific applications, such as crop protection and the production of agricultural inputs [5]. The use of nanoformulations, nanoencapsulates, and nanomaterials will provide us with the next generation of fertilizers and pesticides, allowing for a controlled and accurate application of active ingredients, reducing the action of external agents that cause losses due to leaching, runoffs, and volatilization [6,7]. Furthermore, the quantity of chemical products added to plants and soil will diminish, reducing the environmental impact [8].

Silver nanoparticles (AgNP) exhibit high thermal and electrical conductivity and chemical stability [9]. Additionally, AgNP have been used in various industrial fields as antimicrobial agents [10]. Several publications have demonstrated that silver nanoparticles are efficient against bacteria such as *Bacillus cereus*, *Klebsiella pneumoniae*, and *Enterobacter aerogenes*, and they also exert antifungal effects on *Alternaria alternata*, *Colletotrichum* spp., *Fusarium solani*, *Macrophomina phaseolina*, and *Botrytis cinerea*, among other fungi responsible for causing the diseases that affect a great number of economically important crops [11,12]. However, AgNP, depending on the concentration, exposure time, size, morphology, and soil characteristics, can also affect the beneficial microorganisms present in crops and soils, either native ones or those applied as commercial biopesticides [13,14].

Recent efforts have been developing green technologies for nanoparticle synthesis that do not use toxic substances that generate highly hazardous by-products that can cause environmental pollution [15]. Green synthesis is a feasible alternative that employs biological sources: plants, due to their availability, effectiveness, and low cost [16,17]. One of the plants that have been evaluated for the synthesis of nanoparticles is *Larrea trindentata* (Sapindales: Zygophyllaceae) (Appendix A). Aqueous extracts from *L. trindentata* have proven to be excellent reducers of AgNO_3_. They contain compounds such as phenolic lignans, saponins, flavonoids, amino acids, and minerals, which can substitute toxic compounds in addition to acting as stabilizing agents, allowing the synthesis of functional nanoparticles [18].

In this work, we search for a novel way to overcome multi-drug resistance in pests by using biologically synthesized AgNP, which could represent a suitable option. Previous studies have shown that this kind of NP is less toxic due the residual extract that plays the role of protective capping molecules such as proteins, polyphenols, sugars, alkaloids, and organic acids [19]. However, the application of AgNP in agriculture results in their interaction and their ending up in the ecosystem. Therefore, their toxicology is still unclear, and soil characteristics play a crucial role in transforming these silver nanoparticles into different forms.

The goal of this research was to determine the impact of AgNP synthesized from *Larrea tridentata* (Sessé and Moc. ex DC.) plant extracts on the mycelial growth and morphological changes in beneficial and phytopathogenic fungi, including the following: *Metarhizium anisopliae* (Metschnikoff) (Hypocreales: Clavicipitaceae); *Baeuveria bassiana* (Balsamo) (Hypocreales: Clavicipitaceae); *Trichoderma viridae* (Pers.); and *Trichoderma fasciculatum* Bissett (Hypocreales: *Hypocraceae), as beneficial* strains as well as in *Pestalotia* spp. (Melanconiales: Melanconiaceae); *Colletotrichum gloesporoides* (Pendz) (Glomerellales: Glomerellaceae); *Botrytis cinerea* Persoon (Helotiales: Sclerotiniaceae); *Alternaria solani* Sorauer (Pleosporales: Pleosporaceae); *Macrophomina* spp. (Botryosphaeriales: Botryosphaeriaceae); and *Phytophthora cinnamomi* Rands (Pythiales: Pythiaceae) as pathogenic strains.

## 2. Results

### 2.1. Characterization of Silver Nanoparticles (AgNP)

Figure 1a shows the XRD pattern of the obtained silver nanoparticles (AgNP). The AgNP are composed of the Ag/AgCl phase. The diffraction peaks at 38.2°, 44.2°, 64.5°, and 77.2° (2θ) correspond to the (111), (200), (220), and (311) lattice planes of the FCC structure of Ag (JCPDS PDF 04-0783), and the smaller peaks at 27.8°, 32.2°, 46.2°, 54.9°, and 57.6° correspond to the (111), (200), (220), (311), and (222) lattice planes of the AgCl phase (JCPDS PDF 31-1238). Additionally, in the figure, we can observe the size distributions (Figure 1b) and morphology (Figure 1c) of the synthesized AgNP. The TEM micrograph shows irregular-shaped particles with sizes of 6 to 28 nm.

### 2.2. Antifungal Activity of AgNP

The results showed that the inhibition of the mycelial growth caused by AgNP varies depending on the fungal species. The antifungal activity of the AgNP is different in the fungal strains at 60, 90, and 120 parts per million (ppm). Table 1 shows the inhibition percentage of the radial growth of the pathogenic fungi. The most significant effect was revealed in *A. solani* at 60 ppm, which achieved 70.76 ± 2.68% growth inhibition, followed by *Macrophomina* spp. at 120 ppm with 65.43 ± 7.20%.

On the other hand, *B. cinerea* exhibited the highest inhibition (51.75 ± 3.80%) at the lowest concentration (60 ppm) of AgNP, while *F. oxysporum* reached its maximal growth- inhibition rate (39.04 ± 0.76%). In the case of *P. cinnamomic*, significant differences were observed, but the inhibition did not exceed 23.08%. *Pestolatia* spp. was not sensitive to the AgNP, and it shows a better growth than the control. Similarly, *C. gloesporoides*, at 90 ppm presented the same growth rate as the control, and the inhibition percentages were low (8.57% and 2.86%) compared to the other doses evaluated.

In the case of the beneficial fungi, the strains were less sensitive to the AgNP concentrations (Table 2). *B. bassiana* and *M. anisople* at 60 ppm AgNP did not inhibit the growth of the mycelium, and at 120 ppm, both reached less than 20% of inhibition, while for *T. viridae* and *T. fasciculatum*, the highest inhibition effect was observed at 120 ppm (34.21 ± 0.00 and 30.26 ± 18.28%, respectively). Figure 2 and Figure 3 show the mycelial growth of pathogenic and beneficial fungi at different AgNP concentration rates.

### 2.3. Morphometric Characterization of the Fungi

Figure 4 depicts the mycelia and the structures of *A. solani* and *Macrophomina* spp., the phytopathogenic fungi that experienced the highest levels of growth inhibition by the AgNP (60 and 120 ppm). Hyphal damage was observed under AgNP treatment: the results showed loss of definition, turgor, and constriction sites, as well as the formation of poorly defined mycelial agglomerations.

Regarding beneficial fungi, *T. fasciculatum* and *T. viridae* exhibited spore scattering after being treated with AgNP at 120 ppm, whereas *M. anisopliae* did not reveal any clear morphological or structural changes. In the case of *B. bassiana*, only the treatment with AgNP at 120 ppm produced lower mycelial growth. These changes might be related to the lower numbers and smaller sizes of the spores due to the effects of this compound (Figure 5).

## 3. Discussion

### 3.1. Antifungal Activity of AgNP

According to the observed results, the AgNP could have inhibited the fungal growth depending on the average size of the nanoparticles. The inhibition percentages reported in *A. solani* reached between 100% and 53.7% at concentrations of 10 and 100 ppm, respectively, and in sizes ranging from 7 to 25 nm [20]. Regarding *Macrophomina phaseolina*, prior inhibition reports demonstrated 53.7% mycelial growth inhibition with spherical-shaped AgNP and sizes ranging from 6 to 15 nm at a concentration rate of 100 ppm [21]. In addition, [22] with spherical-shaped bio-AgNP with sizes ranging from 32–47 nm, potential antifungal efficacy was observed, inhibiting the radial growth of *M. phaseolina* by 100% at a concentration of 100 ppm. AgNP biosynthesized from *L. tridentata* extracts had better results, which could be related to the plant extract covering the NP. However, for other strains, such as *B.* cinerea, the results showed less fungal activity compared to those of [23], who reported an inhibition percentage of 52.9% for *B. cinerea* at 15 mg L^−1^.

In the case of *F. oxysporum,* it was not as sensitive to AgNP at the doses evaluated; however, other authors have reported good results. Nonetheless, chitosan-coated AgNP (Ag-CsNP) against *F. oxysporum* at concentrations of 1000, 1500, and 2000 ppm achieved fungal growth reductions of 64%, 70%, and 74%, respectively [24], with chitosan increasing the antifungal effect, similar to the case of *P. cinnamomi*, in which the percentage of inhibition did not exceed 23.08%. However, other authors reported promising results with the combined action of chitosan–silver oligochitosan (OCAgNP) at 9 ppm, resulting in a mycelial-growth inhibition of 79% [25].

The potential of AgNP, synthesized with wormwood aqueous extract (Artemisia absinthium), against Phytophthora parasitica, Phytophthora infestans, Phytophthora palmivora, Phyrophthora cinnamomi, Phytophthora tropicalis, Phytophthora capsicum, and Phytophthora katsurae, on microtitration plates with 10 µg mL^−^¹ of AgNP (10 ppm), achieved 100% efficacy in inhibiting mycelial growth, zoospore germination, the elongation of the germ tube, and zoospores production [26]. The latter indicates that the synthesis of the nanoparticles has a great impact in their antimicrobial activities.

In the case of *Pestolatia* spp., the inhibition effect of AgNP on mycelial growth was scarcely evident; in fact, there was more growth than in the control treatment. This might be due to that certain fungi having properties that enable them to tolerate the toxic action of metals. These properties could include water-proof pigments in their cell walls, extra-cellular polysaccharides, metabolite excretion, complex formation, crystallization, biosorption, bioaccumulation, biomineralization, bioreduction, bio-oxidation, extra-cellular precipitation, biotransformation, and metal-ion output [27]. These mechanisms can explain the reasons why the nanoparticles in this study did not inhibit the mycelial growth of the assessed fungi.

On the other hand, *C. gloesporoides* showed low inhibition of mycelial growth, and no significant differences were observed between the different concentrations of AgNP. These results differ from those reported by other authors. Lamsal et al. [28] evaluated the effect of colloidal-shaped AgNP, with sizes ranging between 4 and 9 nm, in six species of *Colletotrichum.* This group of researchers reported that inhibition increased as the concentration of nanoparticles increased, achieving full growth inhibition at 100 ppm in the strains of *C. acutatum* and *C. gloeosporioides.* Likewise, Aguilar-Méndez et al. [29] reported the fungicidal action of spherical AgNP, with sizes ranging from 5 to 24 nm against *C. gloesporioides,* achieving nearly 90% of growth inhibition at 56 ppm.

In contrast, in the case of beneficial fungi, in neither of the two cases did mycelial growth inhibition exceed 35%. Quite probably, these fungi are tolerant to AgNP, since it has been reported that some fungi can produce macroconidia with complex cell organizations that impair AgNP transportation, in addition to other defense mechanisms [11].

The application of nanotechnology in agriculture is considered one of the promising approaches to increase yield more efficiently [30]. However, nanoparticles can harm beneficial microorganisms by penetrating and deforming fungal hyphae [31]. Furthermore, AgNP likely affects all biological systems, given that their relative toxicity is unknown, as well as the physicochemical parameters and environmental conditions that affect their stability and activity [32]. Therefore, more research is needed to help us to assess the potential impact of nanomaterials on the functioning and sustainability of natural and agricultural ecosystems [31].

### 3.2. Morphometric Characterization of Fungi

The morphological changes observed in the fungi appearance could be related to the cell death of the hyphae. Abdel-Hafez et al. [20] reported the formation of holes and pores in *A. solani* hyphae that were treated with AgNP. The mechanism of action of AgNP is not yet clear. However, some authors have reported that the adverse effect caused by AgNP in fungi is due to morphological, structural, and physiological changes, as well as to alterations in fungal membranes, hyphae, and conidia [33]. Furthermore, they can produce reactive oxygen species (ROS) and can increase permeability of fungal membranes, affecting their functions due to the loss of cell structure and the osmotic imbalance [23,34,35]. Additionally, AgNP cause damage to proteins, lipids, and nucleic acids [23]. In addition, it has been mentioned that the antimicrobial activity of AgNP might be mediated by the formation of free radicals, and free radicals can cause severe damages to the chemical structure of DNA and proteins [36].

The effects of AgNP depends on their shape, size, spherical surface, solubility, superficial load, organic compounds, and their agglomeration status [30,37]. Accordingly, AgNP exhibit the shape-dependent efficacy of bactericidal activities, and their toxic effect increases with reductions in size by having a larger contact surface area [38]. In this research work, the AgNP ranged in size from 2 to 25 nm, which is sufficient for fungal control.

Damage to fungi caused by silver nanoparticles depends on multiple factors. Muñoz-Silva et al. [39] studied the tolerance to heavy metals of *F. oxisporum,* and the results showed a very high tolerance to heavy metals such as Ag, Ag, Pb, Cd, Cr, Cu, Zn and Ni, which, may explain the tolerance of *F. oxisporum* found in this research. In addition, according to Ouda [23], silver nanoparticles caused damage to fungal hyphae, conidia, and cell wall components of *A. solani* and *B. cinérea*. Furthermore, other authors have reported that fungal hyphae showed the formation of pits and pores against different pathogenic isolates of the same *A. solani* fungus after being treated with silver nanoparticles [20]. In addition, Bayat et al. [40] reported that AgNP could successfully inhibit spore germination in *B. cinerea* in a concentration-dependent manner. AgNP application on *Phytophthora parasitica* and *P. capsici* caused a significat inhibition of the on-zoospore germination and germ tube elongation [26]. On other fungi, such as *Fusarium* y *Macrophomina,* the possibility of inhibiting fungal growth could be the results of the production of free radicals by the AgNP [41]. Other effects of AgNP on fungi include deformities in the mycelial growth and the shape of the hyphal walls, with the layers of hyphal walls being torn, and many being collapsed in *Colletothrichum* species [28].

In this research work, the effect of AgNP on cellular components or free radicals was not evaluated; therefore, further studies are suggested to explain the effects of AgNP on the fungi evaluated. However, regarding the effects that have already been reported, we can add loss of definition and turgor, constriction sites, with the formation of poorly defined mycelial agglomerations and lower numbers, and smaller spore size as the resulting effects of this compound.

## 4. Material and Methods

### 4.1. Plant Material

Fresh leaves were collected from plants in the flowering phase in Saltillo, Coahuila, Mexico (25°13′03.7″ N, 101°03′48.7″ W, altitude 2198 masl). A voucher specimen (2022-47B) was pressed and deposited in the herbarium of the Center for Sustainable Development (CEDESU) of the Autonomous University of Campeche (Universidad Autonoma de Campeche, Campeche, Mexico). The following taxonomic characteristics were corroborated: erect woody shrub; ultimate branches appressed-pubescent; leaves bifoliate and densely appressed-pubescent to glabrate; leaflets entire, divaricate, obliquely lanceolate to falcate, and stipules acuminate; sepals appressed-pubescent, 3–4 × 5–8 mm; stamens 5–9 mm; mericarps pilose-woolly; seeds with triangular contour or with a boomerang shape and brown; transverse section of the seeds elliptical or obovate (Appendix A) [42,43].

### 4.2. Silver Nanoparticles (AgNP)

Silver nanoparticles (AgNP) were obtained according to the methodology proposed by [18]. The nanoparticle characterization was carried out through powder X-ray diffraction, in a Rigaku Ultima IV diffractometer using radiation CuKα (λ = 1.5418 Å) at 40 kV and 44 mA, at a scanning speed of 0.02 (2θ/s). The size and morphology of the AgNP were studied via the transmission electron microscopy technique (TEM), using an FEI Titan 80–300 kV microscope.

### 4.3. Sources of Phytopathogenic and Beneficial Fungi

Plant material infected with phytopathogenic fungi was collected from different municipalities of the State of Michoacán. *Alternaria solani*, *Fusarium oxisporum* and *Phytophthora cinnamomic* fungi were isolated from avocado plantations in Uruapan; Colletotrichum gloesporoides was isolated from mango fruits inApatzingan; and *Botrytis cinérea*, *Pestalotia* sp. and *Macrophomina* sp. were obtained from strawberry crops in Zamora. Fungi were taxonomically identified based on the morphology of their structures (Appendix A) [44,45,46,47]. This was examined at 40× magnification under a compound binocular microscope (Carl ZEISS Axio Scope A1). However, the lack of molecular confirmation of the fungi is a limitation of this study.

The entomopathogenic fungi were obtained from commercial products: *Baeuveria bassiana* (Meta-Hiper^®^), *Metarhiziun anisopliae* (Phyto-control^®^), *Trichoderma viridae* and *T. fasciculatum,* which were donated by the plant nutrition laboratory of the Centro de Innovación y Desarrollo Agroalimentario de Michoacán (CIDAM).

### 4.4. Antifungal Activity of AgNP

The Potato-Dextrose-Agar (PDA) culture medium was prepared, sterilized in an autoclave at 121 °C for 15 min, and cooled at 42 °C. The AgNP were added to the medium and placed in Petri dishes. The assessed concentrations were 0, 60, 90 and 120 ppm. A PDA disk of 0.5 cm in diameter, was placed in the center of each Petri dish, with mycelial growth taking place over three days from different fungi. The dishes were incubated at 28 °C, for 6 to 12 days. Each treatment had three replicates.

### 4.5. Mycelial Growth Inhibition

The diameter of the growth of each fungus was measured by digital image analysis using ImageJ^®^ software, Version 1.8.0_172, programmed in Java and developed by the National Institutes of Health (public software). A digital image of the back side of every Petri dish was taken in JPG format, using a mobile phone, Huawei Y9^®^, model JKM-LX3, with a resolution of 4160 × 3120 px.

The mycelial growth was measured in every Petri dish after the control reached the border of the Petri dish, except for *Alternaria solani*, which was measured after 12 days; eight days after incubation for *Beauveria bassiana* and *Metarhizium anisopliae*; and six days after incubation for *Trichoderma viridae* and *T. fasciculatum*. The percentage of mycelial growth inhibition (PICM) was calculated using the following equation proposed by [48]:Inhibition% = (DCC − DCP)/DCC × 100(1)
where: DCC: diameter of the control colony (cm) and DCP: diameter of the problem colony (fungus in the presence of AgNP) (cm).

### 4.6. Morphometric Characterization of Fungi

In order to observe the morphological changes associated with the effects of AgNP over the hyphae and spores of the assessed fungi, pictures of the phytopathogenic fungi showing stronger effects caused by the AgNP, including the beneficial fungi, were taken a with 1000× resolution. We used a fieldeEmission scanning electron microscope, model JSM-7600F, from the Instituto de Investigación en Metalurgia y Materiales at Universidad Michoacana de San Nicolás de Hidalgo.

### 4.7. Data Analysis

Variance analysis and Duncan’s mean comparison test (α = 0.05) were conducted on the percentage of mycelial growth inhibition (PICM). The analysis was performed with the statistical package SAS, Version 8.1 for Windows^®^.

## 5. Conclusions

The antimicrobial effects of AgNP vary according to the species and the microbial strain; however, AgNP can represent an alternative for the management and control of diseases caused by fungi such as *A. solani* and *Macrophomina* spp. due to their inhibiting effect on the mycelial growth of both fungi and their low impact on beneficial fungi (*M. anisopliae*, *B. bassiana*, *T. viridae*, and *T. fasciculatum*). In terms of AgNP changes in fungal morphology (hyphae and spores), more detailed studies are necessary. It is also important to conduct research on diseased plants directly in order to confirm the effectiveness of AgNP and to assess their residual potential effects on beneficial microbiota in order to ensure that the release of silver nanoparticles in agricultural systems does not threaten the sustainability of the environment and the diversity of beneficial microorganism populations. Therefore, further studies will be needed to fully understand the agro-nanotechnological potentialities of AgNP from *L. tridentata.*

## Figures and Tables

**Figure 1 molecules-27-08147-f001:**
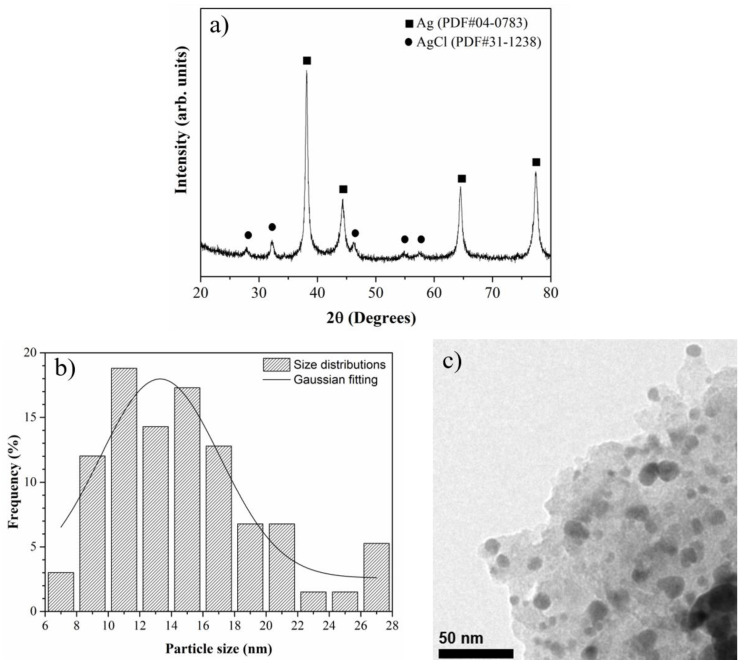
Characterization of AgNP: (**a**) X ray and B ray diffraction patterns, (**b**) micrography obtained in TEM (transmission electron microscopy), (**c**) histogram of AgNP size distribution.

**Figure 2 molecules-27-08147-f002:**
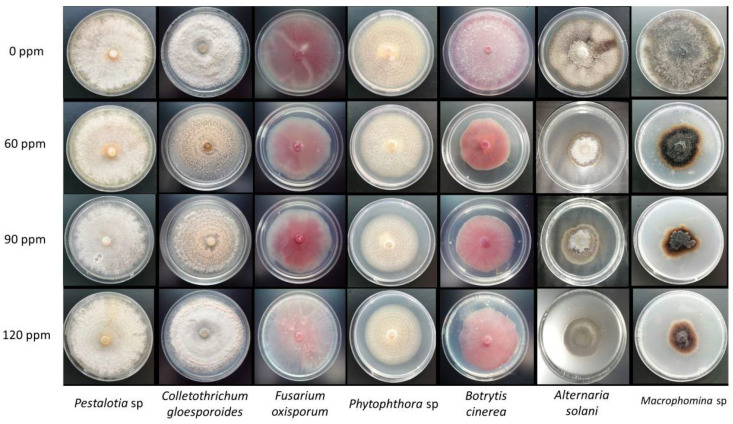
Mycelial growth of phytopathogenic fungi at different AgNP concentration rates, synthesized from the extract of *L. tridentata*.

**Figure 3 molecules-27-08147-f003:**
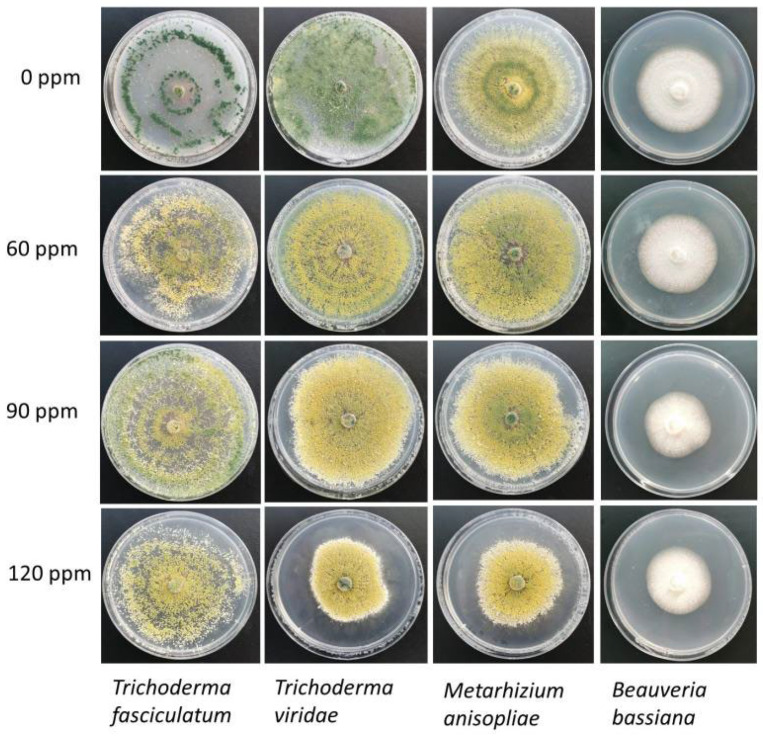
Mycelial growth of beneficial fungi at different AgNP concentration rates, synthesized from the extract of *L. tridentata*.

**Figure 4 molecules-27-08147-f004:**
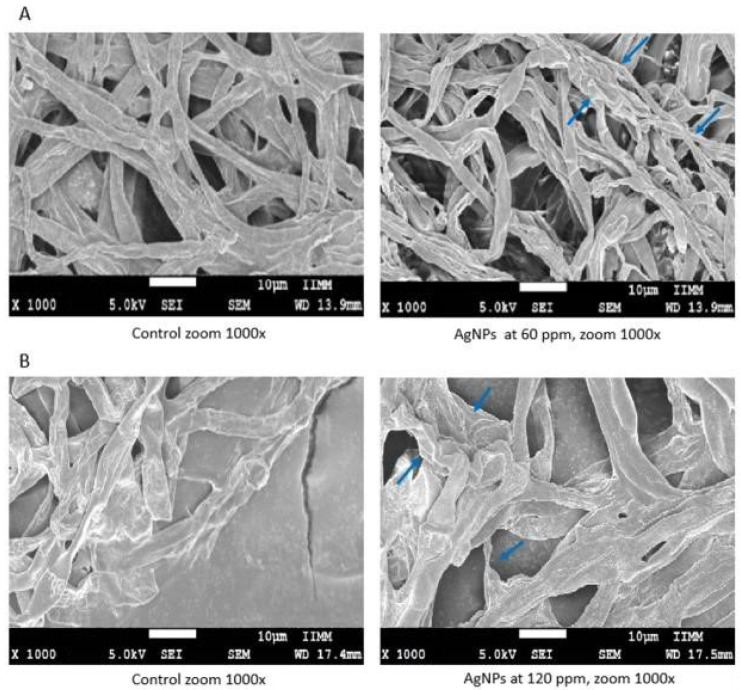
Mycelium under the electron microscope 1000×: (**A**) *Alternaria solani* and (**B**) *Macrophomina* sp. The arrow mark in the image indicates the injury in the fungi structure.

**Figure 5 molecules-27-08147-f005:**
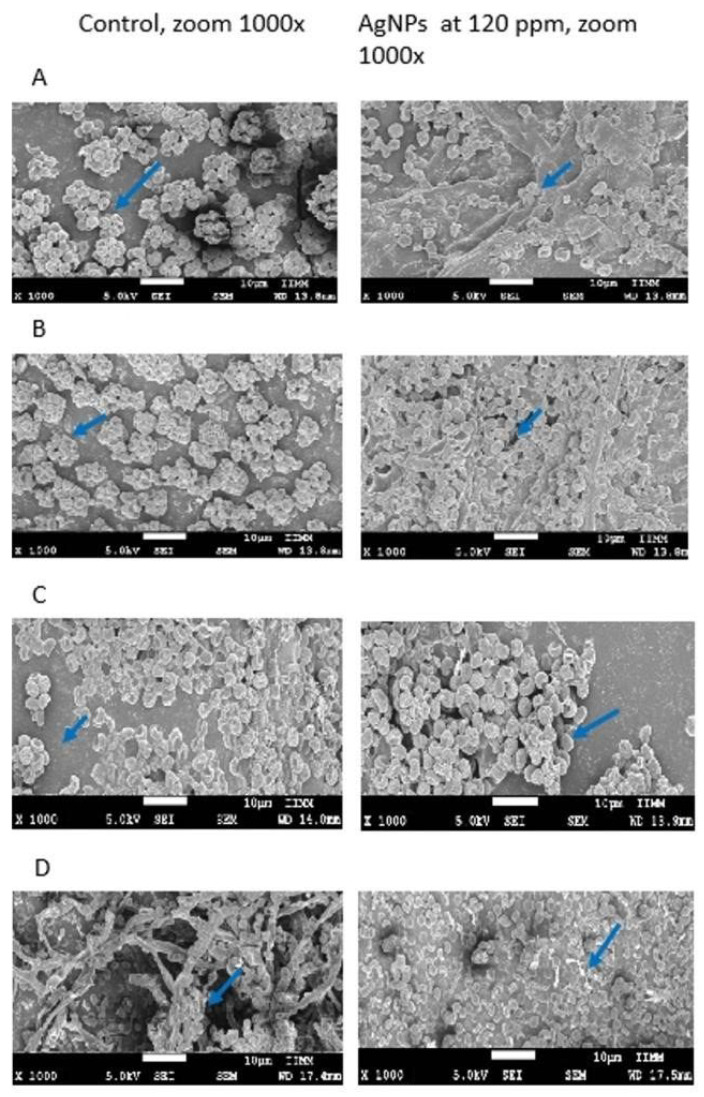
Spores and mycelia of beneficial fungi under the electron microscope at 1000×; (**A**) *Trichoderma fasciculatum*, (**B**) *T. viridae*, (**C**) *Metarhizium anisopliae* and (**D**) *Beauveria bassiana*. The arrow mark in the image indicates the injury in the fungi structure.

**Table 1 molecules-27-08147-t001:** Percentage inhibition of radial growth of phytopathogenic fungi strains at different silver nanoparticles concentrations *.

Strain	Mycelial Growth Inhibition (%)		
	60 ppm	90 ppm	120 ppm
*Alternaria solani*	70.76 ± 2.68 ^a^	60.82 ± 7.09 ^b^	60.23 ± 2.03 ^b^
*Macrophomina* sp.	48.94 ± 7.31 ^b^	64.89 ± 0.00 ^a^	65.43 ± 7.20 ^a^
*Botrytis cinerea*	51.75 ± 3.80 ^a^	46.05 ± 2.28 ^b^	43.86 ± 1.52 ^b^
*Fusarium oxisporum*	39.04 ± 0.76 ^a^	37.28 ± 11.94 ^a^	21.05 ± 3.95 ^b^
*Collethotrichum gloesporoides*	8.57 ± 2.47 ^a^	0.00 ± 3.78 ^ab^	2.86 ± 2.47 ^ab^
*Phytophthora cinnamomi*	8.97 ± 1.11 ^c^	23.08 ± 0.00 ^a^	21.15 ± 0.00 ^b^
*Pestalotia* sp.	−2.70 ± 0.00 ^b^	12.61 ± 8.69 ^a^	11.71 ± 7.80 ^a^

* Eight days after incubation. Means with the same letter in every row did not show significant difference (α = 0.05).

**Table 2 molecules-27-08147-t002:** Percentage inhibition of radial growth of beneficial fungi strains at different silver nanoparticles concentration rates *.

Strain	Mycelial Growth Inhibition (%)		
	60 ppm	90 ppm	120 ppm
*Beauveria bassiana*	0.00 ± 2.17 ^b^	16.25 ± 2.17 ^a^	15.00 ± 2.17 ^a^
*Metarhizium anisopliae*	0.00 ± 0.00 ^a^	5.70 ± 5.93 ^a^	17.98 ± 25.53 ^a^
*Trichoderma viridae*	12.28 ± 0.76 ^b^	14.04 ± 4.98 ^b^	34.21 ± 0.00 ^a^
*Trichoderma fasciculatum*	25.44 ± 0.76 ^a^	1.75 ± 3.04 ^b^	30.26 ± 18.28 ^a^

* Eight days after incubation for *Beauveria bassiana and Metarhizium anisopliae*; and six days after incubation for *Trichoderma viridae and T. fasciculatum*. Means with the same letter in every row did not show significant difference (α = 0.05).

## Data Availability

All data generated or analyzed during this study are included in this published article (and its Appendix A).

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
