# Peer review of "Inhibition of Phytopathogenic and Beneficial Fungi Applying Silver Nanoparticles In Vitro"

_molecules, 2022, doi:10.3390/molecules27238147_

Round 1

Reviewer 1 Report

Although the antifungal activity of the biosynthesized nanoparticle is low, the article is written well, and the research is vital for the agricultural field to control fungal plant diseases based on eco-friendly silver nanoparticles.

However, the points raised in the attached file should be addressed.

Author Response

A pdf document with the asnwers is attached.

Reviewer 2 Report

Dear Authors,

The paper looks good in shape, and for sure it can contribute to the researchers of the field. However, I only have one concern that silver nanoparticles can be used in agriculture since their accumulation in soil or water can cause issues. There should be a justification (with words is enough for this study) to claim this if you guys receive a revision you should add that part.

Kind Regards,

Author Response

A pdf document is attached.

Round 2

Reviewer 1 Report

I added my comments in the attached file.

Author Response

We atach a pdf document
